# *Pseudomonas aeruginosa* Infections in Patients with Severe COVID-19 in Intensive Care Units: A Retrospective Study

**DOI:** 10.3390/antibiotics13050390

**Published:** 2024-04-25

**Authors:** Alexandre Baudet, Marie Regad, Sébastien Gibot, Élodie Conrath, Julie Lizon, Béatrice Demoré, Arnaud Florentin

**Affiliations:** 1INSPIIRE, Inserm, Université de Lorraine, F-54000 Nancy, France; marie.regad@univ-lorraine.fr (M.R.); beatrice.demore@univ-lorraine.fr (B.D.); arnaud.florentin@univ-lorraine.fr (A.F.); 2Service d’Odontologie, CHRU-Nancy, F-54000 Nancy, France; 3Département Territorial D’hygiène et Prévention du Risque Infectieux, CHRU-Nancy, F-54000 Nancy, Francej.lizon@chru-nancy.fr (J.L.); 4Faculté de Médecine, Université de Lorraine, F-54505 Vandœuvre-lès-Nancy, France; 5Service de Médecine Intensive et Réanimation, CHRU-Nancy, F-54000 Nancy, France; 6Pharmacie, CHRU-Nancy, F-54000 Nancy, France; 7Faculté de Pharmacie, Université de Lorraine, F-54505 Vandœuvre-lès-Nancy, France

**Keywords:** *Pseudomonas aeruginosa*, COVID-19, hospital-acquired infections, intensive care unit, antimicrobial resistance

## Abstract

Patients hospitalized in ICUs with severe COVID-19 are at risk for developing hospital-acquired infections, especially infections caused by *Pseudomonas aeruginosa*. We aimed to describe the evolution of *P. aeruginosa* infections in ICUs at CHRU-Nancy (France) in patients with severe COVID-19 during the three initial waves of COVID-19. The second aims were to analyze *P. aeruginosa* resistance and to describe the antibiotic treatments. We conducted a retrospective cohort study among adult patients who were hospitalized for acute respiratory distress syndrome due to COVID-19 and who developed a hospital-acquired infection caused by *P. aeruginosa* during their ICU stay. Among the 51 patients included, most were male (90%) with comorbidities (77%), and the first identification of *P. aeruginosa* infection occurred after a median ICU stay of 11 days. Several patients acquired infections with MDR (27%) and XDR (8%) *P. aeruginosa* strains. The agents that strains most commonly exhibited resistance to were penicillin + β-lactamase inhibitors (59%), cephalosporins (42%), monobactams (32%), and carbapenems (27%). Probabilistic antibiotic treatment was prescribed for 49 patients (96%) and was subsequently adapted for 51% of patients after antibiogram and for 33% of patients after noncompliant antibiotic plasma concentration. Hospital-acquired infection is a common and life-threatening complication in critically ill patients. Efforts to minimize the occurrence and improve the treatment of such infections, including infections caused by resistant strains, must be pursued.

## 1. Introduction

*Pseudomonas aeruginosa* is an opportunistic Gram-negative bacterium that is largely associated with hospital-acquired infections (HAIs) [1,2]. These infections can lead to ventilator-associated pneumonia (VAP), bloodstream infection (BSI), catheter-associated urinary tract infection (cUTI), and surgical infection [1]. Worldwide, *P. aeruginosa* is the fifth most common microorganism responsible for bacterial infection, and it causes the most deaths and years of life lost in hospitals [3]. The 30-day mortality for *P. aeruginosa* bacteraemia is approximately 40% [4]. *P. aeruginosa* is one of the main bacteria responsible for HAIs [2], especially in intensive care units (ICUs), where 40% of *P. aeruginosa* infections are distributed [5].

A slight increase in the incidence of HAIs caused by *P. aeruginosa* was observed during the COVID-19 pandemic [6]. The risk of acquiring HAIs significantly increases with the severity of COVID-19 and the duration of hospitalization [7]. A large meta-analysis that described 355,579 COVID-19 patients across 138 studies performed between December 2019 and May 2021 revealed that the prevalence of bacterial coinfection and secondary infection among COVID-19 patients in ICUs were 8.4% and 39.9%, respectively [8]. Patients with severe COVID-19 who were hospitalized in ICUs were significantly more likely to be coinfected with bacteria [9,10]. In France, COVID-19 patients in ICUs were significantly more likely to be infected by *P. aeruginosa* bacteraemia (12.4% vs. 8.9%) and VAP (16.8% vs. 16.1%) than those without COVID-19 [11]. Coinfections, including *P. aeruginosa* infections, which occurred among COVID-19 patients hospitalized in ICUs during the three initial waves of the COVID-19 pandemic (from March 2020 to May 2021) had a negative impact on patient prognosis, with a mortality rate of 74% [12]. Mortality increased among hospitalized patients with resistant and multidrug resistant (MDR) *P. aeruginosa* infections compared to those with susceptible *P. aeruginosa* infections [13].

*P. aeruginosa* is complex to treat because it is intrinsically resistant to many antimicrobial agents, and this bacterium can additionally acquire other resistances [14]. Among hospitalized COVID-19 patients, *P. aeruginosa* is among the main bacterial species with resistance to antibiotics, accounting for approximately 25% of resistant strains [15]. In 2020, in Europe, 30.1% and 17.3% of 20,675 invasive *P. aeruginosa* isolates reported to EARS-Net were resistant to one and at least two antimicrobial groups under surveillance (piperacillin/tazobactam, fluoroquinolones, ceftazidime, carbapenems, and aminoglycosides) [14].

During the first waves of the COVID-19 pandemic, the consumption of penicillin with β-lactamase and carbapenems significantly increased in ICUs [16]. The main antibiotics prescribed in hospitalized patients with COVID-19 were amoxicillin/clavulanic acid, piperacillin/tazobactam, cephalosporines, and carbapenems [17]. The inappropriate use of antibiotic treatments was frequently described and could lead to infections by MDR bacteria, which have a significant impact on the management of hospitalized patients with COVID-19 infections, leading to a prolonged hospitalization time and increased mortality [7,8,15,18].

Due to the serious consequences of *P. aeruginosa* infections among hospitalized patients and, more particularly, patients admitted to ICUs with severe COVID-19, additional studies are needed. Moreover, a focus on the antibiotic resistance of *P. aeruginosa* is needed because the WHO has listed carbapenem-resistant *P. aeruginosa* as a pathogen of critical priority that requires further research and development [19]. A recent review indicated that antibiotic resistant *P. aeruginosa* in patients with BSIs was rarely reported in the literature during the COVID-19 pandemic and needs to be further explored [20].

The main aim of this study was to describe the evolution of *P. aeruginosa* infections in patients hospitalized in ICUs at CHRU-Nancy (France) for Acute Respiratory Distress Syndrome (ARDS) due to COVID-19 during the three initial waves of the COVID-19 pandemic in France (1 March 2020 to 31 May 2021). The second aim was to analyze the resistance of *P. aeruginosa* strains that cause infection and to describe the antibiotic treatments.

## 2. Results

### 2.1. Demographic and Clinical Data

A total of 645 adult patients were admitted for COVID-19 in ICUs and continuing care units in our hospital from 1 March 2020 to 31 May 2021. During this period, 185 adult patients admitted in the ICUs in our hospital were infected or colonized by *P. aeruginosa*, mainly males (75%) with a mean age of 62 ± 11 years. Among them, 53 adult patients met the inclusion criteria, but two refused to participate in the study. The characteristics of the 51 patients included in this study are summarized in Table 1. The participants were aged 65 ± 12 years (range: 36–82) with a body mass index of 30 ± 6 kg/m^2^, and 90% were male. Three-quarters of the patients (77%) presented with at least one comorbidity, mainly arterial hypertension (61%), obesity (45%), or chronic cardiac disease (45%). The median hospital stay and ICU stay were 36 (range: 6–196) and 27 days (range: 6–172), respectively. Patients were mainly admitted from smaller peripheral hospitals (59%), and 43% died during their ICU stay. The two main causes of death were multiple-organ failure (55%) and respiratory failure (23%). Only two significant differences between survivors and patients who died were highlighted: the patients who died were older (*p* < 0.03) and the survivors had longer hospital stays (*p* < 0.03).

Prior to hospital admission, it was documented that 41% of patients received at least one antibiotic treatment, mainly beta-lactams (37%)—including third-generation cephalosporins such as cefotaxime and ceftriaxone (26%), amoxicillin/clavulanate (22%), piperacillin/tazobactam (8%), and amoxicillin (4%)—and macrolides (26%) including spiramycin (22%) and azithromycin (4%).

All patients (100%) received oxygenation support during their ICU stay: 22% were treated by extracorporeal membrane oxygenation, and 94% were treated with invasive mechanical ventilation for a median length of 30 days (Q1–Q3: 19–35). The use of prone positioning, corticosteroids, and neuromuscular blockers was prescribed for most patients (86%, 69%, and 61%, respectively).

### 2.2. Pseudomonas aeruginosa Infections

Among the 51 patients, 15 acquired their first *P. aeruginosa* infection during the first COVID-19 pandemic wave, 12 during the second, and 24 during the third (Figure 1).

During these three waves, the median delay between the first diagnosis of COVID-19 and the identification of a *P. aeruginosa* infection among patients was 14 days (Q1–Q3: 10–25). The first identification of a *P. aeruginosa* infection occurred after a median ICU stay of 11 days (Q1–Q3: 6–18). After one month (30 days) in the ICU, 90% of patients were infected with *P. aeruginosa,* and 24% were infected with MDR *P. aeruginosa* (Figure 2).

A total of 188 strains of *P. aeruginosa* were collected from the patients. After removing duplicate strains, 122 *P. aeruginosa* strains obtained from 106 samples and comprising 118 antibiograms were included in this study. The four strains without antibiograms corresponded to respiratory samples that were analyzed by multiplex polymerase chain reaction (m-PCR; FilmArray^®^).

Among the 51 patients, *P. aeruginosa* was mainly responsible for VAP (*n* = 47; 92%), BSIs (*n* = 7; 14%), and catheter-associated BSIs (*n* = 5; 10%); eight patients developed *P. aeruginosa* infections from several of these sites (VAP + BSIs: *n* = 4; VAP + catheter-associated BSIs: *n* = 2; VAP + BSIs + catheter-associated BSIs; *n* = 2). Regarding VAP, *P. aeruginosa* was isolated from patients via endotracheal aspiration (*n* = 39; 76%), sputum (*n* = 5; 10%), bronchoalveolar lavage (*n* = 5; 10%), and a telescoping plugged catheter (*n* = 5; 10%). Regarding BSIs, only one patient presented exclusively a BSI without relapse, and the six others presented BSIs simultaneously or subsequently to VAP by *P. aeruginosa* with an identical antibiogram.

Twenty-five (49%) patients acquired more than one *P. aeruginosa* infection during their ICU stay, with 22 (43%) patients infected by several strains, 12 (24%) infected at several sites, and three (6%) patients with relapsed infections.

Regarding antibiotic resistance, 20 patients (39%) were exclusively infected by susceptible strains, 14 (27%) were infected by MDR strains, and four (8%) were infected by XDR strains; among them, three (6%) were infected by difficult-to-treat resistant (DTR) strains. The 118 antibiograms included 39 (33%) susceptible strains, 35 (30%) MDR strains, and eight (7%) XDR strains (Table 2); among them, six (5%) were DTR strains (0 first strain, 6 subsequent strains). The first strains sampled were significantly more susceptible (*p* < 0.0001) and had less MDR (*p* < 0.01) and less DTR (*p* < 0.04) than the subsequent strains (i.e., strains other than the first among patients infected by several *P. aeruginosa* strains).

Regarding the 118 antibiograms performed, the main agents that strains were resistant to were penicillin + β-lactamase inhibitors (*n* = 70; 59%), cephalosporins (*n* = 50; 42%), monobactams (*n* = 38; 32%), and carbapenems (*n* = 32; 27%) (Table 3). Regarding these four antimicrobial categories, the first strains sampled had significantly less resistance than the subsequent strains (*p* ≤ 0.002). Among the 22 patients infected with several strains of *P. aeruginosa*, 20 (91%) had subsequent strains that were more resistant.

Thirty-six (71%) patients acquired coinfections with *P. aeruginosa*, and 58 of 106 samples (55%) contained other microorganisms in addition to *P. aeruginosa*. The three main species identified were *Candida* sp. (*n* = 40), *Klebsiella* sp. (*n* = 12), and *Stenotrophomonas maltophilia* (*n* = 7).

### 2.3. Antibiotic Treatments

All patients (100%) were treated with antibiotics during their ICU stay for a median of 20 days (Q1–Q3: 12–27), representing approximately 70% of their ICU stay.

The ten most prescribed antibiotics are presented in Table 4; they represented 85% of the total antibiotic prescriptions. The median prescription duration was ≤7 days for each antibiotic, and they were always prescribed intravenously. A total of 194 plasma dosages were used for these ten antibiotics in 39 (76%) patients: 99 (51%) of the dosages were compliant, 53 (27%) were an overdosage, and 42 (22%) were an underdosage.

For the first *P. aeruginosa* infection, probabilistic antibiotic treatment was prescribed for 49 patients (96%); the agents used included piperacillin-tazobactam (*n* = 30, 58%), meropenem (*n* = 6, 12%), cefepime (*n* = 5, 10%), ceftazidime (*n* = 4, 8%), and imipenem (*n* = 4, 8%). After obtaining the antibiogram, 26 patients (51%) had an adaptation of their antibiotic treatment within 72 h; they were treated with antibiotics, such as ceftazidime (*n* = 18, 35%), cefepime (*n* = 4, 8%), imipenem (*n* = 3, 6%), and ciprofloxacin (*n* = 1, 2%). A total of 31 antibiotic treatments were administered with at least one antibiotic plasma dosage to 24 (47%) patients. The first antibiotic plasma dosages were compliant in 13 (42%) patients, overdosed in 11 (35%) patients, and underdosed in seven (23%) patients. The antibiotic treatment dosage was adapted in five (45%) overdosage cases and one (14%) underdosage case.

Among the seven patients who acquired BSIs by *P. aeruginosa*, four were initially treated by ineffective antibiotics (antibiograms revealed resistance to antibiotics previously prescribed), two had initiated their antibiotic treatments < 48 h before the blood samples, one was not treated by antibiotic before the blood sample, and one presented a BSI with a susceptible strain despite 10 days of piperacillin-tazobactam. After BSI identification and antibiogram results, all patients were treated with susceptible antibiotics for *P. aeruginosa*.

## 3. Discussion

In our hospital, 53 adult patients acquired *P. aeruginosa* infection in the ICU after admission for ARDS due to COVID-19 during the three initial waves of the COVID-19 pandemic. During this period, numerous patients infected with COVID-19 presented with ARDS and were hospitalized in ICUs [21]. Up to 56% of these patients acquired bacterial coinfections, mainly VAP, BSIs, and UTIs, which often led to sepsis and septic shock [22,23]. *P. aeruginosa* infections were mainly identified in respiratory samples and from male patients [5]. Our results are in accordance with these previous studies [5,22,23] because, among the 51 included patients, 90% were males, and *P. aeruginosa* was mainly responsible for VAP (92%), BSIs (14%), and catheter-associated BSIs (10%).

Appropriate management of COVID-19 patients who develop HAIs in the ICU is crucial because these infections can occur in numerous patients, lead to difficulties in treatment, and have a high mortality rate (43% of patients who acquired *P. aeruginosa* died in our study).

Several clinical conditions may explain the increased rate of HAIs during the COVID-19 pandemic, including prolonged hospitalization in the ICU (the median ICU stay was 27 days in our study, with 50% of patients affected by *P. aeruginosa* after 11 days), worse clinical presentation at ICU admission (all the included patients in our study presented with ARDS, all received oxygenation support, and 84% required prone positioning), the use of systemic corticosteroid treatment for ARDS in the ICU (69% of patients were treated with corticosteroids in our study), the presence of immunosuppressive comorbidities such as diabetes (25% of patients had diabetes in our study), the administration of immune-modulator treatments such as tocilizumab, and the use of invasive mechanical ventilation (94% of patients were treated with invasive mechanical ventilation for a median duration of 30 days in our study) [7,24]. One major explanation of the association between the duration of the ICU stays and the acquisition of *P. aeruginosa* HAIs is the exposure to at-risk invasive devices such as mechanical ventilation, vascular catheters, and urinary catheters [25].

Regarding antibiotic resistance, 61% of the patients included in our study were infected with resistant strains of *P. aeruginosa*, 27% with MDR strains and 8% with XDR strains. The resistance profiles of the first strains of *P. aeruginosa* identified in our study were comparable to those of *P. aeruginosa* strains in France in 2020: piperacillin-tazobactam resistance (13% vs. 17%), ceftazidime resistance (11% vs. 13%), carbapenem resistance (11% vs. 13%), aminoglycoside resistance (6% vs. 6%), and MDR (14% vs. 8%) [14]. The subsequent strains sampled in our study were significantly more resistant, with the selection of resistant *P. aeruginosa* strains and the acquisition of resistances.

*P. aeruginosa* is not the only concerning bacteria; all the ESKAPE pathogens (*Enterococcus faecium*, *Staphylococcus aureus*, *Klebsiella pneumoniae*, *Acinetobacter baumannii*, *P. aeruginosa*, and *Enterobacter* spp.) are causative bacteria of HAIs well-known to present antibiotic resistances [26]. The main Gram-negative ESKAPE bacteria isolated from COVID-19 patients were *A. baumannii*, *K. pneumoniae*, and *P. aeruginosa* [27,28]. Most of the studies highlighted an increase in the prevalence of carbapenem-resistant *A. baumannii* and *K. pneumoniae* [18]. The highest prevalence of resistance was reported for MDR *A. baumannii* [8], with a prevalence from 40 to 95% in patients with HAI pneumonia and a mortality from 45 to 85% [26].

In their narrative review, Gaudet et al. reported that critically ill COVID-19 patients presented with a greater incidence of HAIs with MDR isolates in the ICU. These authors highlighted several pathophysiological factors that could explain these findings, including COVID-19-mediated post-aggressive immunoparalysis; exposure to antibiotics, steroids, and other immunomodulating agents; a longer ICU stay; exposure to invasive devices; and organizational constraints triggered by the pandemic [29]. During the first waves of the COVID-19 pandemic, nurses in the ICU had an excessive workload; this increase in workload was associated with poor quality of care, unfavourable patient safety, and unfavourable infection prevention [30]. Moreover, the lack of adequate personal protective equipment for frontline health care workers (including respirators, gloves, face shields, gowns, and hand sanitizers) was reported, potentially heightening the risk of MDR transmission [31].

Antibiotic exposure is a significant risk factor for the spread of antimicrobial resistance [8]. Indeed, the use of antibiotics exerts selective pressure that isolates the resistant strains, particularly for longer treatments [32]. A study performed in 260 United Kingdom hospitals that included nearly 49,000 hospitalized COVID-19 patients revealed that 37% of patients received antibiotics prior to hospital admission, and 85% received antibiotics during hospitalization [33]. In our study, 41% of patients received antibiotics before hospital admission; this percentage could be underestimated due to the retrospective collection of data reported by patients at admission to our hospital. All included patients were treated with antibiotics during their ICU stay for a median of 20 days. The use of antibiotics in ICUs is crucial for treating patients infected with bacteria such as *P. aeruginosa*. However, the proper use of antibiotics is essential for fighting antibiotic resistance and reducing the side effects of treatment [34]. Unfortunately, we highlighted the fact that probabilistic antibiotic treatment was prescribed for 96% of patients and was only adapted for 51% of patients after receiving an antibiogram and 33% of patients after receiving a noncompliant antibiotic plasma dosage. These percentages of adaptation of the antibiotic treatments could be underestimated due to the retrospective collection of data from electronic patient records; ICU teams were able to quickly adapt the treatment but only reported this change a few days later in the hospital software. To improve antibiotic prescription and to adapt treatments with relevant clinical and microbial information in real time (such as antibiogram and antibiotic plasma dosage results), our hospital installed a computerized decision support system for antimicrobial stewardship after the COVID-19 pandemic [35]. This initiative was implemented to improve antibiotic treatments and the management of HAIs in a similar context to increase the quality of care.

This study describes the in-depth characteristics of patients admitted to a French hospital for ARDS due to COVID-19 during the three initial waves of the COVID-19 pandemic and who acquired *P. aeruginosa* infections during their ICU stay. Moreover, this study precisely describes the resistance of *P. aeruginosa* and provides new insights into the increase in the antibiotic resistance of strains during the patients’ ICU stay. The two main limitations of this study are linked to its design because it was an observational retrospective monocentric study with a limited number of patients, which may have affected the analysis and generalizability of the results. On the one hand, the retrospective collection of data from electronic patient records could underestimate several outcomes such as the percentage of patients receiving antibiotics before hospital admission and the percentages of adaptation of antibiotic treatments after antibiogram and after antibiotic plasma dosage. This limitation is frequent in this field of research; Langford et al. reported in their systematic review that among 148 studies regarding antimicrobial resistance in patients with COVID-19, all were observational, and most were retrospective cohort studies (81%) [8]. On the other hand, despite the inclusion of patients hospitalized in several ICUs in different buildings of our hospital with different medical teams, this study includes a small number of patients in a single hospital. So, this study presented a limited external validity, limiting the generalization of the data.

## 4. Materials and Methods

### 4.1. Study Design and Setting

We conducted a single retrospective cohort study in ICUs at CHRU-Nancy (Regional University Hospital of Nancy) during the three initial waves of the COVID-19 pandemic in France. The first wave occurred from 1 March to 1 August 2020, the second from 2 August 2020 to 1 January 2021, and the third from 2 January to 31 May 2021. During the initial waves of the COVID-19 pandemic, the CHRU-Nancy received critical COVID-19 patients in two ICUs and two continuing care units, which were upgraded to ICUs (up to 48 beds), and in one surgical ICU and two surgical continuing care units.

Ethics approval was obtained from the Ethics Committee of CHRU-Nancy (approval no. 413). The study was registered on 21 November 2023 on ClinicalTrials.gov (Identifier: NCT06141837).

### 4.2. Patient Selection

The inclusion criteria were as follows:Adult patients were hospitalized for at least 48 h in an ICU at CHRU-Nancy for ARDS due to COVID-19 (confirmed by reverse transcription polymerase chain reaction or an antigenic test).Patients were hospitalized from 1 March 2020 to 31 May 2021.Patients developed a HAI caused by *P. aeruginosa* during their ICU stay.

The exclusion criteria were as follows:Patients < 18 years old.Patients without COVID-19 at the ICU admission.Patients with *P. aeruginosa* isolated <48 h following ICU admission.Patients who did not want to be included in the study.

### 4.3. Data Collection and Outcomes

Epidemiological and demographic data, medical history, and comorbidities at the time of hospitalization were reported. Data on antibiotic treatments received before hospital admission, the length of hospital and ICU stay, and mortality were also obtained directly from medical records. The records of all patients with positive *P. aeruginosa* results were reviewed, and infections occurring 48 h after hospital admission were categorized as BSI, VAP, or cUTI according to CDC definitions. The bacterial susceptibility of *P. aeruginosa* was interpreted according to the European Committee on Antimicrobial Susceptibility Testing (EUCAST) criteria [36].

As described by Lyu et al. [5], duplicate samples were eliminated, and when multiple strains were isolated from the same site, with the same antibiogram and from one patient within one month, they were designated as belonging to the same strain, with the first isolate being used as a representative sample.

*P. aeruginosa* relapse was defined as a positive culture within 30 days after discontinuing antibiotic therapy for the initial infection.

Following the *P. aeruginosa* categories listed by Magiorakos et al. [37], bacteria were classified as multidrug resistant (MDR) if they acquired nonsusceptibility to at least one agent in three or more antimicrobial categories, extensively drug resistant (XDR) if they acquired nonsusceptibility to at least one agent in all but one or two antimicrobial categories, and pandrug resistant (PDR) if they acquired nonsusceptibility to all agents in all antimicrobial categories. According to Kadri et al. [38], difficult-to-treat resistant (DTR) *P. aeruginosa* was defined as intermediate or resistant to all reported agents in carbapenems (meropenem and imipenem), β-lactam (ticarcillin-clavulanic acid, piperacillin-tazobactam, ceftazidime, cefepime, and aztreonam), and fluoroquinolone (ciprofloxacin) categories.

To describe the evolution of *P. aeruginosa* infections among patients, the number of newly infected patients was reported each month. To describe the resistance and susceptibility of *P. aeruginosa* strains, antibiograms following the EUCAST criteria were used [36]. The French guidelines [39,40,41] were used to analyze the antibiotic treatments, the appropriateness of the prescriptions (molecule, dose, route of administration, and duration), and the compliance of the antibiotic plasma dosages.

### 4.4. Statistical Analysis

The data were collected using Microsoft Excel and analyzed using RStudio (R version 4.3.2). The data are presented as numbers and percentages for categorical variables and as the means ± standard deviations (SDs) and ranges (min–max) or medians and interquartile ranges (Q1–Q3) depending on the distribution of continuous variables. Chi-squared tests were used for categorical variable analysis or Fisher’s exact tests when the expected frequencies were less than five, and the Bonferroni correction was used for multiple comparisons. The Mann–Whitney U test was used for continuous variable analysis. The statistical significance was set at *p* < 0.05.

## 5. Conclusions

Patients who were hospitalized in ICUs for severe COVID-19 during the initial waves of the COVID-19 pandemic and who developed *P. aeruginosa* infections had difficult-to-treat infections and high mortality. The acquisition and increase in antibiotic resistance among *P. aeruginosa* strains during the ICU stay complicated treatment. Efforts to minimize the likelihood of acquiring such infections, especially infections with MDR, XDR and DTR *P. aeruginosa*, must be pursued. Adaptation of antibiotic treatments according to antibiograms and antibiotic plasma dosage results should be quickly performed in infected patients to optimize antimicrobial treatments.

## Figures and Tables

**Figure 1 antibiotics-13-00390-f001:**
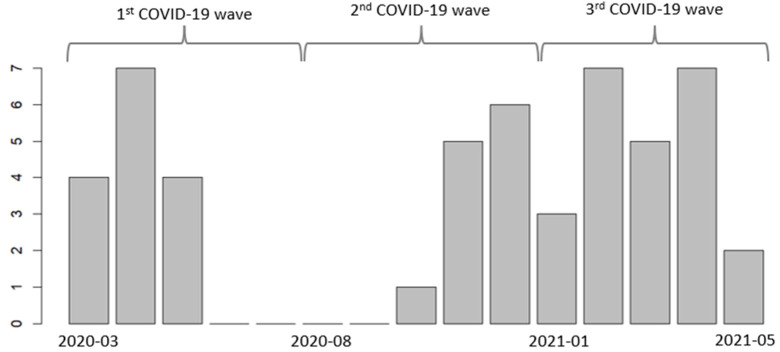
The number of first identified *P. aeruginosa* infections each month among 51 patients admitted to the intensive care unit (ICU) for ARDS due to COVID-19 during the three initial waves.

**Figure 2 antibiotics-13-00390-f002:**
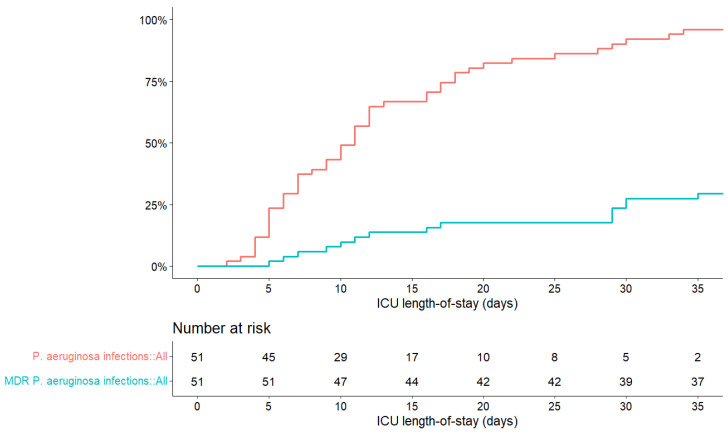
Occurrence of *P. aeruginosa* infections (red) and multidrug resistant (MDR) *P. aeruginosa* infections (blue) in 51 patients in the intensive care unit (ICU).

**Table 1 antibiotics-13-00390-t001:** Demographic and clinical characteristics of COVID-19 patients infected with *Pseudomonas aeruginosa* during their ICU stay and data based on survival during their hospital stay.

Patient Characteristics	All Patients (*N* = 51)	Survivors (*n* = 29)	Patients Who Died (*n* = 22)
**Demographic characteristics**			
Age (years)	65 ± 12	62 ± 12	69 ± 11
BMI (kg/m^2^)	30 ± 6	30 ± 6	29 ± 7
Gender (male)	46 (90%)	25 (86%)	21 (95%)
**Comorbidities**			
Arterial hypertension	31 (61%)	19 (63%)	12 (55%)
Obesity (BMI ≥ 30)	23 (45%)	15 (52%)	8 (40%)
Chronic cardiac disease	23 (45%)	13 (42%)	10 (46%)
Kidney insufficiency	20 (39%)	9 (29%)	11 (50%)
Obstructive pulmonary disease *	15 (29%)	10 (33%)	5 (23%)
Diabetes	13 (25%)	8 (27%)	5 (23%)
Malignancy	3 (6%)	2 (7%)	1 (5%)
**Hospital stays**			
Hospital length-of-stay (days)	36 (27–62)	42 (32–69)	31 (21–42)
ICU length-of-stay (days)	27 (20–39)	27 (20–40)	27 (21–36)
Admission from another hospital	30 (59%)	15 (52%)	15 (68%)
Admission from home/emergency	21 (41%)	14 (48%)	7 (32%)
Death during stay	22 (43%)	0 (0%)	22 (100%)
Discharge to home	14 (27%)	14 (48%)	0 (0%)
Discharge to another hospital	8 (16%)	8 (28%)	0 (0%)
Discharge to recuperative care center	7 (14%)	7 (24%)	0 (0%)
**Treatments**			
Antibiotics before hospital admission	21 (41%)	9 (31%)	12 (55%)
Antibiotics during ICU stay	51 (100%)	29 (100%)	22 (100%)
Prone position	44 (86%)	25 (86%)	19 (86%)
Corticosteroids	35 (69%)	18 (62%)	17 (77%)
Neuromuscular blockers	31 (61%)	16 (55%)	15 (68%)
**Maximal oxygenation support**			
Extracorporeal membrane oxygenation	11 (22%)	7 (24%)	4 (18%)
Invasive mechanical ventilation	37 (72%)	19 (66%)	18 (82%)
Non-invasive ventilation	2 (4%)	2 (7%)	0 (0%)
High-flow oxygen	1 (2%)	1 (3%)	0 (0%)

The data are expressed as the mean ± SD, median (Q1–Q3), or *n* (%). * Obstructive pulmonary disease including chronic obstructive pulmonary disease and obstructive sleep apnea; BMI: body mass index; ICU: intensive care unit.

**Table 2 antibiotics-13-00390-t002:** Antibiotic resistance of the 118 *Pseudomonas aeruginosa* antibiograms among 51 patients.

Antibiotic Resistance	All Strains (*N* = 118)	First Strains (*n* = 51)	Subsequent Strains (*n* = 67)	*p*-Value
Susceptible	39 (33%)	33 (65%)	6 (9%)	<0.0001
Resistant *	36 (31%)	10 (19%)	26 (39%)	0.11
MDR	35 (29%)	7 (14%)	28 (42%)	<0.01
XDR	8 (7%)	1 (2%)	7 (10%)	0.54
PDR	0 (0%)	0 (0%)	0 (0%)	-

* Resistant to one or two antimicrobial categories; MDR: multidrug resistant; XDR: extensively drug resistant; PDR: pandrug resistant.

**Table 3 antibiotics-13-00390-t003:** Antibiotic resistance profiles of 118 *Pseudomonas aeruginosa* antibiograms from 51 patients.

Antimicrobial Category Antimicrobial Agent	All Strains (*N* = 118)	First Strains (*n* = 51)	Subsequent Strains (*n* = 67)	*p*-Value
**Penicillin** **+ β-lactamase inhibitors**	70 (59%)	13 (25%)	57 (86%)	<0.0001
Ticarcillin-clavulanic acid	70 (59%)	13 (25%)	57 (86%)	<0.0001
Piperacillin-tazobactam	61 (52%)	7 (13%)	54 (82%)	<0.0001
**Cephalosporins**	50 (42%)	7 (13%)	43 (65%)	<0.0001
Ceftazidime	49 (42%)	6 (11%)	43 (65%)	<0.0001
Cefepime	26 (22%)	6 (11%)	20 (30%)	0.03
**Monobactams**	38 (32%)	8 (15%)	30 (45%)	0.002
Aztreonam	38 (32%)	8 (15%)	30 (45%)	0.002
**Carbapenems**	32 (27%)	6 (11%)	26 (39%)	0.001
Meropenem	31 (26%)	6 (11%)	26 (39%)	<0.001
Imipenem	29 (25%)	5 (9%)	24 (36%)	0.002
**Cephalosporins ** **+ β-lactamase inhibitors**	21 (18%)	5 (9%)	16 (24%)	0.08
Ceftazidime-avibactam	19 (16%)	3 (6%)	16 (24%)	0.02
Ceftolozane-tazobactam	15 (13%)	4 (8%)	11 (17%)	0.27
**Fluoroquinolones**	13 (11%)	4 (8%)	9 (14%)	0.51
Ciprofloxacin	13 (11%)	4 (8%)	9 (14%)	0.51
**Aminoglycosides**	5 (4%)	3 (6%)	2 (3%)	0.75
Gentamicin	5 (4%)	3 (6%)	2 (3%)	0.75
Tobramycin	1 (1%)	1 (2%)	0 (0%)	0.90
Amikacin	0 (0%)	0 (0%)	0 (0%)	-

**Table 4 antibiotics-13-00390-t004:** The ten most commonly prescribed antibiotics among the 51 patients during their ICU stay.

Molecule	Patients *n* (%)	Duration in Days Median (Q1–Q3)	Plasma Dosage Compliance *n*/*N* (%)	Plasma Dosage in mg/L Median (Q1–Q3)
Piperacillin-tazobactam	39 (76%)	4 (2–7)	17/41 * (41%)	83.9 (42.4–125.2)
Cefotaxime	33 (65%)	5 (3–7)	2/6 (33%)	21.5 (17.7–24.9)
Ceftazidime	25 (49%)	7 (5–10)	18/31 (58%)	62.0 (44.6–86.0)
Amikacin	24 (47%)	1 (1–1)	13/28 (46%)	41.6 (4.7–84.2)
Cefepime	21 (41%)	6 (3–9)	5/15 (33%)	42.9 (33.6–64.6)
Spiramycin	20 (39%)	4 (3–5)	0/0 (NA)	NA
Meropenem	18 (35%)	4 (3–11)	9/26 (35%)	15.9 (10.4–39.7)
Imipenem	13 (25%)	3 (2–7)	0/7 (0%)	7.2 (2.0–15.1)
Vancomycin	9 (18%)	5 (4–7)	35/40 (88%)	27.3 (22.0–33.8)
Linezolid	8 (1%)	4 (2–7)	0/0 (NA)	NA

* Only piperacillin was dosed; NA: not applicable.

## Data Availability

The data presented in this study are available upon reasonable request to the corresponding authors.

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
