# Peer review of "Pseudomonas aeruginosa Infections in Patients with Severe COVID-19 in Intensive Care Units: A Retrospective Study"

_antibiotics, 2024, doi:10.3390/antibiotics13050390_

Round 1
Reviewer 1 Report
Comments and Suggestions for Authors
The study conducted by Baudet A et al. focuses on Pseudomonas aeruginosa infections in patients with severe COVID-19 hospitalized in intensive care units (ICUs). Conducted at CHRU-Nancy, France, it retrospectively analyzed cases during the first three waves of the COVID-19 pandemic. The primary objectives were to track the evolution of P. aeruginosa infections and examine antibiotic resistance patterns and treatment strategies. The study highlights the significant burden of P. aeruginosa infections and the challenges in managing antibiotic resistance in severe COVID-19 cases in ICUs. The paper is very well-written and well-organized. However, the study's retrospective design may limit the ability to establish causality between observed factors and outcomes.
(1) The study includes a relatively small sample size, which might not represent the broader population of ICU patients with COVID-19.
(2) Please include the discussion of this limitation, specifically the study's single-center design and its potential impact on generalizability, in the discussion section of the manuscript.
(3) Without a comparative non-COVID-19 ICU patient group, it's challenging to isolate the impact of COVID-19 on the incidence and severity of P. aeruginosa infections.
(4) Detailed microbiological characterization of P. aeruginosa strains, including genotyping, could provide insights into transmission dynamics and resistance mechanisms.
(5) The study primarily focuses on the presence of P. aeruginosa and resistance patterns, with less emphasis on patient outcomes such as mortality or length of ICU stay directly attributed to these infections.
(6) The study lacks follow-up data on patient outcomes post-ICU discharge, which could provide valuable insights into the long-term impact of P. aeruginosa infections.
Author Response
See attachement

Reviewer 2 Report
Comments and Suggestions for Authors
Line 37: A reference is missing after (HAIs)
Line 70-71: specify which antibiotics were frequently used during COVID-19
Line 69-74: the link between MDR bacteria and the presence of Pseudomonas aeruginosa/immune-suppressive treatments is not clear. you need to reword this paragraph.
Line 158: 118 instead of 119
Line 192: A reference is missing
Line 198: A reference is missing
Line 197-207: Explain why the frequency of Pseudomonas aeruginosa infections increases when hospitalisation is prolonged. And how these infections can occur.
Line 228: This sentence is not clear. You need to rephrase it.
Author Response
See attachement

Round 2
Reviewer 1 Report
Comments and Suggestions for Authors
The authos have addressed my concerns and I accept their explanations.